# Can 16 Minutes of HIIT Improve Attentional Resources in Young Students?

**DOI:** 10.3390/jfmk8030116

**Published:** 2023-08-11

**Authors:** Karina E. Andrade-Lara, Pedro Ángel Latorre Román, Juan Antonio Párraga Montilla, José Carlos Cabrera Linares

**Affiliations:** Department of Musical, Plastic and Corporal Expression, University of Jaén, 23071 Jaén, Spain; karinandrade9011@gmail.com (K.E.A.-L.); platorre@ujaen.es (P.Á.L.R.); jccabrer@ujaen.es (J.C.C.L.)

**Keywords:** physical education, attention, cognitive function, HIIT, visuoperceptive ability

## Abstract

Attentional resources are a cornerstone of both cognitive and academic performance. The purpose of this study was to analyse the effect of high-intensity interval training (HIIT) sessions on selective attention and visuoperceptual ability in young students. A total of 134 students (12.83 ± 1.23 years) joined this study. They were randomly assigned to a control group (CG) (*n* = 67), which watched a documentary, or an experimental group (EG) (*n* = 67), which performed 16 min of HIIT. Attention and visuoperceptual ability were assessed through the Perception of Similarities and Differences test (Caras-R test). A repeated-measures two-way ANOVA analysis was conducted. The CG showed an increased number of errors compared to the EG (*p* < 0.001) and showed a lower Impulsivity Control Index (*p* < 0.001) after the investigation. The EG, meanwhile, showed an increased number of hits (*p* < 0.001), Impulsivity Control Index (*p* < 0.001), and attentional efficacy (*p* < 0.001). In addition, the EG showed a decreased number of errors (*p* < 0.001) and omissions (*p* < 0.01). In conclusion, 16 min of HIIT was time-effective in improving selective attention and visuoperceptual ability in young students. These results show the importance of physical exercise and the promotion of physical activity breaks during the academic day to improve learning processes.

## 1. Introduction

The regular practice of physical activity (PA) in adolescence represents improvements not only in physical condition, cardiorespiratory fitness, agility, strength, coordination, flexibility [1], and mental health [2], but also improvements in executive function and academic performance in children and adolescents [3].

Executive function represents the set of higher-level cognitive abilities whose function is to initiate, adapt, regulate, supervise, and control the information from different mental and behavioural processes and adaptation to the environment [4]. In this sense, adolescence can be seen as a sensitive neurobiological period that is characterised by increased brain plasticity, which means more development opportunities; however, this age is also associated with greater brain vulnerability [5]. In particular, selective attention is the process through which a person is able to select and focus on specific information necessary for the fulfilment of a task, while alluding to inconsequential information or distractors, both internal (own thoughts or automatic responses) and external (auditory, visual, or environmental stimuli), that hinder the specific execution of a task [6]. On the other hand, attentional efficacy (AE) is the cognitive capacity that a person possesses to regulate and optimise the attentional process to select a stimulus and maintain attention over a certain period of time [7,8]. Visual perception (VP), meanwhile, is the ability to interpret visual information received from the environment and give meaning to what is observed [9]. Consequently, visuoperceptive abilities in children and adolescents contribute to the progress of academic performance and daily activities [10].

In the last decade, scientific evidence has shown the benefits of regularly practising moderate to vigorous PA both at a cognitive [11] and physiological [12,13] level. One of the modalities that has caused the greatest interest has been high-intensity interval training (HIIT) compared to other training programmes. HIIT is a method that combines brief bursts of exercises executed at high intensity, interspersed with short periods of active recovery at low intensity [14]. Martin-Smith et al. [15] concluded in their research that HIIT produces improvements in cardiorespiratory fitness and metabolic parameters in adolescents.

In addition to the aforementioned effects, this methodology produces improvements at the cognitive level, since it has an influence on selective attention, attentional breadth (which is defined as greater attentional focus at a global level, which implies the greater extension of visual attention) [16], and visuoperceptive ability [7,17]. HIIT promotes cerebral synaptogenesis, increases levels of brain-derived neurotrophic factor (BDNF), produces greater cerebral blood flow, and stimulates the process of neurogenesis [18]. Furthermore, earlier studies reported the benefits of high-intensity training as a useful tool in improving some cognitive skills, such as school aptitudes, creativity, and attention and cognitive flexibility [19,20,21], as well as cognitive outcomes and academic performance [22]. A narrative review carried out by Tomporowski et al. [23] concluded that improving mental functioning characteristics through physical exercise may be a key strategy to develop cognitive growth. In this regard, Pesce et al. [24] concluded that applying two types of exercise (aerobic circuit training versus team games) improved memory performance in physical education classes. It may also assist in providing greater memory storage in children due to an acute bout of submaximal exercise. Furthermore, Tomporowski et al. [25] reported that components of information processing and cognitive function in adults are facilitated by submaximal aerobic exercise undertaken for periods up to 60 min. Pontifex et al. [26] concluded that bouts of acute aerobic exercise may improve cognition in young people.

Best [27] reported that exercise, both short- and long-term, may support executive function in children. The basis for this theory is the fact that aerobic exercise produces not only general but also specific physiological changes in the brain, besides causing an immediate neurochemical response that may enhance cognitive performance. 

However, long-term training programmes in PA, followed consistently, are more likely to enhance executive function than a single PA session [22], although recent research has concluded that short bursts of intense aerobic exercise may be utilised to activate cognitive resources while studying in class, which is advantageous for academic performance [27,28]. Nowadays, teachers complain that there has been a decrease in children’s attention span and the increased development of attention deficits. Likewise, Mahar et al. [29] concluded that long stretches of learning without a break may be detrimental to academic achievement. Previous research [30,31] highlighted how primary-school-aged children who participate in extended academic teaching sessions frequently report increased restlessness as well as decreased concentration and attention. As a result, worse academic performance is linked to attention deficiencies [32].

Nevertheless, despite the influence of HIIT on cognitive function, the existing literature has not examined the effects of submaximal aerobic exercise on students’ attentional resources. Gallotta et al. [33] demonstrated that the effects of high-intensity aerobic exercise on cognitive function were changed or influenced by the exercise duration, intensity, physical fitness, and the particular physical activity conducted during exercise, indicating that high-intensity aerobic exercise appears to be a better at boosting cognitive functioning. Some authors, in fact, have highlighted the impact of high-intensity aerobic exercise in increasing executive function [34,35,36,37]. Moreover, Du Rietz et al. [38] concluded that 20 min of high-intensity exercise enhanced cognition and attentional resources in adults.

Despite the benefit of physical exercise in improving both children’s cognitive outcomes and their academic performance, there is a gap in the scientific literature regarding the effects of HIIT on attentional resources in adolescents. Therefore, the purpose of this study was to analyse the effect of HIIT sessions on selective attention and visuoperceptual abilities in young students.

## 2. Materials and Methods

### 2.1. Design and Participants

A total of 134 students (12.83 ± 1.23 years) took part in this quasi-experimental study, of which 75 were boys (age= 12.88 ± 1.45 years) and 56 were girls (age = 12.84 ± 0.92). The participants were randomly assigned to a control group (CG) (*n* = 67; girls: *n*= 28; boys: *n* = 29) and an experimental group (EG) (*n* = 67; girls: *n* = 29; boys: *n* = 28) through the cluster random sampling technique. The inclusion criteria were (a) being enrolled during the 2020–2021 school year; (b) presenting an informed consent form signed by their parents or legal tutor; (c) not having intellectual or physical disabilities (this criterion was based on the psychological school team). In addition, the principles of the Helsinki Declaration (Helsinki, 2013) were followed. This study was approved by the Bioethics Committee of the University of Jaén (JUN.21/7.TES).

### 2.2. Instruments

The evaluation of selective attention was carried out through the Perception of Similarities and Differences test (Caras-R test) developed by Thurston and Yela [39]. In terms of reliability, this test obtained a Cronbach’s alpha (α) level of 0.92. The test assesses selective attention, sustained attention, and the ability to quickly perceive differences in partially ordered stimulant patterns. It is structured by 60 graphic elements (blocks—stimuli) grouped into three schematic drawings of faces (eyes, eyebrows, hair, mouth), in which two faces are the same and one is different. The test consists of crossing out the face that is different from the other two from the same blocks of faces. The time required for the execution of the test is three minutes. The maximal score of the test is 60 points, which is obtained from the sum of the hits, mistakes, and omissions. Moreover, the Impulsivity Control Index (ICI) and the attentional efficacy were obtained. The ICI indicates whether participants are impulsive in making decisions or discriminating objectives in a specific time. This index was calculated through the following equation: (hits − mistakes)/(hits + mistakes). Meanwhile, the AE was obtained as follows: AE = hits/(hits + mistakes + omissions). The score obtained in the EA is a value between 0 and 1, which represents the probability of hitting each time the individual makes an attempt. Thus, the value 1 indicates that the subject is 100% capable of marking all the stimuli correctly, without incurring any error [40].

The weekly PA level was obtained through the Moderate–Vigorous Physical Activity (MVPA) questionnaire in its original version [41]. The instrument comprises two items that record the number of days for which PA is performed in a usual and atypical week, estimating a duration of 60 min of moderate to vigorous PA. In addition, Borg’s Rating of Perceived Exertion (RPE) was used to evaluate the level of workout intensity. Borg’s scale [42] is a tool that measures a person’s exertion, dyspnoea, and fatigue during physical work. This scale ranges from 0 (low intensity) to 10 (high intensity).

### 2.3. Procedure

The application of the instruments was carried out during the first hour of the academic day (8 a.m.). Firstly, the Caras-R test and the MVPA questionnaire were applied. After this, the intervention was carried out in both groups.

HIIT intervention. The activity performed by the EG consisted of the execution of a HIIT session. The initial part of the session began with a five-minute standard warming-up period (skipping, butt kicks, knee-to-chest lifts, jogging exercises, and dynamic stretching). The main part of the HIIT session was developed over 16 min, structured by four sets of four exercises per set. Each exercise lasted for 1 min. The participants performed 30 s of work and 30 s of rest per exercise (i.e., a work ratio of 1:1). Sets 1 and 3 were structured by the exercises: jumping jacks, lunges, backs, star jumps, and sprints (20 m). The exercises executed in sets 2 and 4, meanwhile, were mountain climbers, skaters, high knees, and running burpees. The session was run by a physical education specialist using the HIIT method, alongside the supervision of the researcher and the student’s tutor. The intensity of the HIIT session was registered using Borg’s scale. Prior to performing the session, the children were trained to understand and perform exercises with a value over 6 on the scale.

Non-HIIT intervention. The activity carried out by the CG consisted of watching a documentary called “The Science of Obesity” (Nat Geo, 2011). The selection of the topic was based on the work that was being carried out in the school period by the student’s tutor, which concerned raising awareness about healthy habits. Once both interventions had been completed, the Caras-R post-test was applied again.

### 2.4. Statistical Analysis

The results were expressed as means and standard deviations (SD) in the case of quantitative variables and as absolute frequencies (%) for qualitative variables. The one-way ANOVA statistic was used to determine the differences in each group. In addition, the chi-square test was used to compare the active and inactive children. A two-way repeated-measures ANOVA (control vs. experimental group) was carried out at two time points (pre-test and post-test) to analyse the effect of the HIIT session. A 95% confidence interval was considered and a significance level of *p* ≤ 0.05 was established. Statistical analyses were conducted using the statistical software IBM SPSS^®^ Statistics 25.0 for Windows (SPSS Inc., Chicago, IL, USA).

## 3. Results

Table 1 shows the sociodemographic characteristics of the participants. With regard to the PA results, significant differences were observed in PA practice in the last week (*p* < 0.05), in a usual week (*p* < 0.05), and in average moderate to vigorous PA (*p* < 0.05). Comparing the groups with respect to the categorisation of the participants according to the level of PA, the CG presented higher percentages of inactive children than the EG (*p* < 0.001).

Figure 1 shows the differences found in the pre and post measures between groups. The results of the two-way repeated-measures ANOVA showed an significant interaction of time × group in the following variables: hits—F (1.132) = 194.98; *p* ≤ 0.001; mistakes—F(1.132) = 24.71; *p* ≤ 0.001; omissions—F(1.132) = 83.92; *p* ≤ 0.001; ICI—F(1.132) = 62.71; *p* ≤ 0.001, and AE—F(1.132) = 195.16; *p* ≤ 0.001. A more detailed analysis between the groups for each measure (pre and post) showed that, in the pre-omission measures (*p* < 0.05), the groups showed significant differences. On the other hand, in the post measurements between the groups, higher means were observed in the EG in the hits and AE variables (*p* < 0.001, respectively). Furthermore, ICI showed significant differences (*p* < 0.001). Note that a reduction in the number of mistakes (*p* < 0.05) and omissions (*p* < 0.001) with respect to the CG was found.

Table 2 presents the two-way ANOVA statistical analysis for repeated measures carried out on the variables of the Caras-R test, with the CG showing a reduction in AE after the intervention (*p* < 0.01). On the other hand, the EG showed a significant increase in the number of hits (*p* < 0.001) and a reduction in the number of mistakes and omissions (*p* > 0.001), while, in the ICI and AE, an increase was observed (*p* < 0.001) after the intervention. With respect to exercise intensity using the Borg scale, the EG showed a greater perception (7.52 ± 1.54) of the physical work intensity level than the CG (1.04 ± 1.05) after the intervention (*p* < 0.001).

## 4. Discussion

The objective of this study was to analyse the effect of a HIIT session on selective attention and visuoperceptual ability in young students. The main findings obtained showed that the HIIT session had a positive effect on perceptual–attentional skills, since the EG participants showed an increased total number of correct answers, ICI, and AE, and a significantly decreased total number of errors and omissions in the Caras-R test after the HIIT session.

These findings are in line with previous research [12,19,21] corroborating the effects of moderate to vigorous PA practice on brain activity in children and adolescents, thereby becoming an efficient mechanism in increasing brain function and cognition. In addition, HIIT has been shown to be an effective tool in increasing attention, concentration, memory, and involvement in tasks, and in improving other executive functions that contribute to enhancing academic and cognitive performance at different stages of life [43].

These results are also corroborated by previous research that evaluated attention in a single PA session in children and adolescents, demonstrating significant relationships between the measures analysed [44]. Similarly, Rosa-Guillamón et al. [40] showed in their study that the participants (48 schoolchildren), after completing a HIIT programme, showed significant improvements in the hits, errors, omissions, ICI, and AE variables, corroborating the results obtained in this research, despite the fact that the sample used comprised preadolescents. Another investigation carried out by the same researcher [45] showed that, after walking a mile in the shortest possible time at moderate to vigorous intensity, significant improvements in attention were evidenced since the students improved in the variables of the Caras-R test (hits, omissions, and ICI) after the intervention.

In addition, the research carried out by Ma et al. [46] showed that four minutes of high-intensity interval training significantly improved selective attention in students. The efficacy of HIIT training on attention and other executive functions in children and adolescents is due to the activation of cognitive function. This activation is induced by high-intensity aerobic exercise, through increased cerebral blood flow, increased brain grey matter, neurogenesis, and other brain functions that affect cognition and academic performance [47]. Moreover, Du Rietz et al. [38] concluded in their study that 20 min of high-intensity exercise improves cognition and attentional function. Kucab et al. [48] explained that HIIT has a positive effect on academic performance and suggested that it should be practised during break times, since it helps to mitigate declines in cognitive performance throughout the academic day.

However, results contrary to those obtained in this investigation were observed in the investigations of Zervas et al. [49], who did not find significant differences in cognitive variables after individual training in high-intensity running on a treadmill over 25 weeks in physical education classes. Moreover, Wilson et al. [50] concluded that after 10 min of intervention with active breaks at a moderate to vigorous intensity in 58 schoolchildren, no significant differences were observed in students’ attention. In the same way, Perciavalle et al. [51], in their research, concluded that selective attention decreased (with increased errors and omissions) due to the presence of lactate in the blood after a CrossFit session.

Finally, the line of research on the study of PA in attention and other executive functions in the academic performance of children and young people has been increasing since the different scientific evidence in this field has allowed us to recognise the importance of PA in cognitive processes throughout human life, but, above all, they have contributed in the academic field to an increase in the number of hours of physical education, which is why it is important to continue with the research process in this field.

This study had some limitations that need to be highlighted: (a) due to the heterogeneity of the participants, as well as the difficulty in obtaining a larger sample, it was not possible to perform an analysis by gender in which information on the effects of HIIT based on gender could have been obtained; (b) longitudinal studies are necessary to support evidence of HIIT and cognitive process; (c) the intensity of HIIT should be monitored using a PA heart rate monitor to better estimate physical exertion. These results must be interpreted with caution because the sample size was limited and therefore not representative of all students in the ages analysed in this study. In the same way, the fluctuations in maturation at these ages can lead to differences among students of the same age and sex, so it is necessary to carry out studies that make it possible to analyse the effects that HIIT has depending on age and sex.

## 5. Conclusions

In conclusion, the results obtained in this study show that a HIIT session has positive effects on attentional resources such as selective attention and visuoperceptual ability in young students. As regards the improvements obtained in attentional resources, this work methodology (HIIT session at the beginning of the school day) could be used by education professionals to increase the attention of their students during the school day. The results emphasise the importance of physical education classes as a tool in promoting the importance of PA as an alternative approach to improve academic performance.

## Figures and Tables

**Figure 1 jfmk-08-00116-f001:**
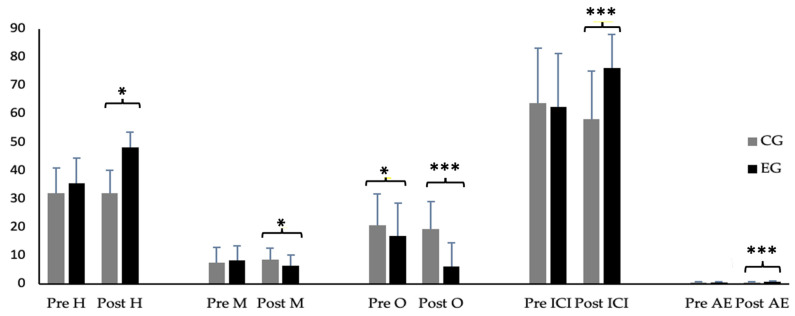
Results of the Caras-R test between groups in the pre and post measurement. * denotes significant differences at * *p* > 0.05; *** *p* > 0.001. H: hits; M: mistakes; O: omissions; ICI: Impulsivity Control Index; AE: attentional effectiveness; CG: control group; EG: experimental group.

**Table 1 jfmk-08-00116-t001:** Sociodemographic characteristics of the participants. Mean (m), standard deviation (SD), and frequency (%).

Variables	All		CG		EG		
	(*n* = 134)		(*n* = 67)		(*n* = 67)		
	Mean	SD	Mean	SD	Mean	SD	*p*-Value
Age (years)	12.83	±1.23	12.97	±0.95	12.72	±1.42	0.234
PA (days/last week)	4.06	±2.06	3.70	±2.05	4.39	±2.02	0.042
PA (atypical days/week)	3.37	±2.21	2.89	±2.08	3.78	±2.25	0.037
Total PA (days)	3.72	±2.04	3.29	±1.96	4.09	±2.04	0.041
PA level	*n*	%	*n*	%	*n*	%	
Inactive	91	63.6	52	77.6	39	51.3	<0.001
Active	52	36.4	15	22.4	37	48.7	

PA: physical activity; CG: control group; EG: experimental group.

**Table 2 jfmk-08-00116-t002:** Results of the Caras-R test between the pre and post measurements in each group.

	CG (*n* = 67)					EG (*n* = 67)				
Variables	Pre		Post		*p*-Value	Pre		Post		*p*-Value
	Mean	SD	Mean	SD		Mean	SD	Mean	SD	
Hits	32.17	(8.64)	31.98	(8.17)	0.898	34.82	(7.81)	49.38	(5.07)	<0.001
Mistakes	7.37	(5.39)	8.65	(4.10)	0.501	9.13	(5.25)	6.89	(3.45)	<0.001
Omissions	20.44	(10.95)	19.43	(9.62)	0.247	16.04	(9.95)	3.71	(5.29)	<0.001
ICI	64.08	(19.07)	58.11	(16.88)	0.003	59.27	(19.40)	75.72	(11.27)	<0.001
AE	0.53	(0.17)	0.52	(0.14)	0.721	0.58	(0.13)	0.82	(0.08)	<0.001

ICI: Impulsivity Control Index; AE: attentional efficacy; CG: Control group; EG: Experimental group.

## Data Availability

Not applicable.

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
