# Peer review of "Can 16 Minutes of HIIT Improve Attentional Resources in Young Students?"

_jfmk, 2023, doi:10.3390/jfmk8030116_

Round 1

Author Response

Thank you for the time you have spent reviewing our manuscript. We have added all the changes that you suggested us since we think that it will improve our manuscript. You can find all our response separated point by point below (in bold) in the document attached. Also, all the changes have been added in the manuscript (highlighted with yellow colour).

Reviewer 2 Report

First of all, I want to thank the journal Editor for the opportunity to review this paper. The research is interesting. There is no doubt that the human body operates as a system, and the relationship between cognitive activity and physical activity has been investigated at all ages, both in young people and the elderly. From this perspective, the study seems interesting, especially concerning the link between high-intensity training and cognitive capacity.

However, there is a major limitation, which is the lack of experimental control. There are many variables that can influence the results, including gender. Therefore, I request the authors to specify the number of boys and girls in both the experimental and control groups in the sample description.

On the other hand, it is necessary to report statistics that I consider crucial to support the results. I suggest that the authors perform the following:

A repeated measures ANOVA, where they report the p-values for 1) within-group differences, 2) between-group differences, and 3) interaction. In other words, a repeated measures ANOVA with two factors (before vs. after) and (EG Vs CG).

Furthermore, they should also provide the mean of the test results. The reason for this is as follows: In Table 2, the variables present a very high standard deviation (SD) in relation to the mean; thus, the existence of some outliers is foreseeable. In my opinion, knowing the median could clarify this from, the median is robust statistic.

Once the authors provide this information, I will be able to review the paper in more detail.

Author Response

(The authors gave the same response as above.)

Round 2

Reviewer 2 Report

In my opinion, this version can be published, after the corrections made by the authors, I consider that the results provide relevant information, although it has important limitations that must be taken into account in future research designs.